# Predictive Factors of Recurrence in Patients with Differentiated Thyroid Carcinoma: A Retrospective Analysis on 579 Patients

**DOI:** 10.3390/cancers11091230

**Published:** 2019-08-22

**Authors:** Fabio Medas, Gian Luigi Canu, Francesco Boi, Maria Letizia Lai, Enrico Erdas, Pietro Giorgio Calò

**Affiliations:** 1Department of Surgical Sciences, University of Cagliari, 09042 Cagliari, Italy; 2Department of Medical Sciences, University of Cagliari, 09042 Cagliari, Italy; 3Division of Pathological, University of Cagliari AOU, 09127 Cagliari, Italy

**Keywords:** differentiated thyroid carcinoma, recurrent carcinoma, lymph node metastasis, microcarcinoma

## Abstract

Differentiated thyroid carcinoma (DTC) is usually associated with a favorable prognosis. Nevertheless, up to 30% of patients present a local or distant recurrence. The aim of this study was to assess the incidence of recurrence after surgery for DTC and to identify predictive factors of recurrence. We included in this retrospective study 579 consecutive patients who underwent thyroidectomy for DTC from 2011 to 2016 at our institution. We observed biochemical or structural recurrent disease in 36 (6.2%) patients; five-year disease-free survival was 94.1%. On univariate analysis, male sex, histotype, lymph node yield, lymph node metastasis, extrathyroidal invasion and multicentricity were associated with significantly higher risk of recurrence, while microcarcinoma was correlated with significantly lower risk of recurrence. On multivariate analysis, only lymph node metastases (OR 4.724, *p* = 0.012) and microcarcinoma (OR 0.328, *p* = 0.034) were detected as independent predictive factors of recurrence. Postoperative management should be individualized and commensurate with the risk of recurrence: Patients with high-risk carcinoma should undergo strict follow-up and aggressive treatment. Furthermore, assessment of the risk should be repeated over time, considering individual response to therapy.

## 1. Introduction

Differentiated thyroid carcinoma (DTC) is the most frequent endocrine tumor, including papillary thyroid carcinoma (PTC), follicular thyroid carcinoma (FTC) and poorly-differentiated thyroid carcinoma; its incidence has been increasing during the last decades [1,2,3,4]. PTC is the most common subtype of DTC, and its incidence has increased by 3% yearly in the US [1]. Primary treatment consists in thyroidectomy, eventually followed by Radioactive Iodine (RAI) ablation. Most of DTC are low-risk tumors, with little or no effect on mortality [2,5,6]. Nevertheless, the incidence of local and distant recurrence is not negligible, reaching 30% in some series [7,8,9]; this event leads to the need of subsequent cycles of RAI therapy and, eventually, of further surgical treatment and chemotherapy, with a reduction in quality of life of patients. Several factors have been described as predictive of recurrence; among these, pathologic features of the tumor have been included in the risk stratification system for structural disease recurrence proposed by the American Thyroid Association. The objective of this system is to identify patients with a significant risk of recurrence, which require postoperative RAI ablation and intensive postoperative surveillance, and, by the other side, to recognize patients at low-risk of recurrence, in which an adequate follow-up is enough.

The aim of this study was to evaluate the recurrence rate of DTC in a high-volume, regional referral center for thyroid carcinoma, and to identify predictive factors of disease recurrence.

## 2. Methods

We included in this retrospective observational study all patients who underwent thyroidectomy at our Department of General and Endocrine Surgery from January 2011 to December 2016 with histopathological diagnosis of DTC. Patients were identified from a prospectively maintained institutional database, and those with incomplete data were excluded from the study. Informed consent was obtained from every patient. Procedures were in accordance with the Helsinki Declaration of 1975, as revised in 1983. Ethical approval for this study was obtained from local ethical committee. During the study period, a total number of 1975 patients with thyroid disease underwent surgical therapy; among them, 579 met the inclusion criteria.

### 2.1. Preoperative Evaluation

All patients included in the study underwent preoperative assessment including physical examination, dosage of serum thyroid hormones and high-resolution ultrasonography (US) of the neck. In case of suspicious nodules, fine needle aspiration cytology (FNAC) was performed. Thyroid scintigraphy was done only in patients with hyperthyroidism, defined as low serum TSH (<0.4 mIU/L) and high or normal serum free T4 and free T3. Preoperative laryngoscopy was routinely performed to assess vocal cords mobility.

### 2.2. Surgical Treatment and Pathologic Examination

All patients underwent extracapsular total thyroidectomy. Intraoperative neuromonitoring of recurrent laryngeal nerve (RLN) was routinely used; RLNs were exposed until their insertion in larynx. Parathyroid glands were routinely researched during dissection, and any attempt to preserve them was made. In patients with preoperative diagnosis or intraoperative suspicion of lymph node metastasis, a therapeutic central compartment lymph node dissection (CLND) or modified radical lateral neck dissection (LND) was performed. According to American Thyroid Association (ATA) and National Comprehensive Cancer Network (NCCN) guidelines, patients with high-risk DTC underwent prophylactic CLND.

Thyroid cancers were classified according to TNM classification of AJCC (7th edition, 2010) [10].

### 2.3. Postoperative Management and Follow-Up

All patients were referred to endocrinologists for postoperative management.

RAI was routinely administrated after total thyroidectomy in ATA intermediate and high-risk patients.

Serum Thyroglobulin (Tg) and Anti-Tg antibodies measurements and neck ultrasound (US) were used in for postoperative evaluation. During initial follow-up, serum Tg and Anti-Tg antibodies were measured every 6–12 months. More frequent Tg and Anti-Tg antibodies measurements were performed in ATA high-risk patients. In ATA low and intermediate-risk patients that achieved an excellent response to treatment, Tg measurements were repeated every 12–24 months. ATA high-risk patients (regardless of response to therapy) and all patients with biochemical incomplete, structural incomplete or indeterminate response to treatment continued to execute Tg and Anti-Tg antibodies measurements at least every 6–12 months for several years.

In patients with DTC of any risk level with significant comorbidity that precluded thyroid hormone withdrawal prior to RAI therapy, recombinant human TSH (rhTSH) preparation was done; these situations included medical or psychiatric conditions that could be acutely exacerbated in case of hypothyroidism (leading to a serious adverse event) or inability to mount an adequate endogenous TSH response with thyroid hormone withdrawal.

Disease-free status was defined as a No Evidence of Disease (NED) and included the following features: no clinical evidence of tumor, no imaging evidence of disease by RAI imaging and/or neck US, and low serum Tg levels during TSH suppression (Tg < 0.2 ng/mL) or after stimulation (Tg < 1 ng/mL) in the absence of interfering antibodies.

Persistent disease was classified as an alteration on US, serum Tg or serum levels of Tg antibodies found within one year after initial operation. Conversely, we defined recurrent disease as clinical, biochemical or structural evidence of disease found one year after surgery or later in a patient considered disease-free after primary treatment. 

### 2.4. Statistical Analysis

The primary endpoint of this study was the recurrence rate of DTC. Univariate analysis was conducted to evaluate the influence of preoperative and pathological factors on recurrence; Chi-squared test and Student’s *t*-test were used for categorical data and for continuous variables, respectively. Variables ≤ 0.100 in the univariate analysis were considered potentially significant and were then included in the multivariate analysis. Logistic regression analysis was used to identify independent risk factors of recurrence. Results were considered statistically significant if p-value was ≤ 0.05. Disease-free survival was defined as the time from initial surgery to the detection of recurrence; log-rank test was used to estimate the statistical differences in Kaplan-Meier curves for independent risk factors. Calculations were performed with MedCalc^®^ vers. 19.0.4. Continuous variables are expressed as mean ± standard deviation of the mean.

## 3. Results

### 3.1. Preoperative Data

We included in the study 579 patients with diagnosis of DTC (Table 1). There were 139 (24%) male and 440 (76%) female, with a mean age of 50.7 ± 14.1 years. Preoperative FNAC was performed in 471 (81.3%) patients. Serum thyroid hormones revealed euthyroidism in 526 (90.8%) cases and hyperthyroidism in 53 (9.2%). Familiarity for thyroid cancer was present in 103 (17.8%) patients, and autoimmune thyroiditis in 215 (21.6%). There were no patients with distant metastases at diagnosis.

Indication to surgery was malignant disease (based on clinical, ultrasonographic and cytological findings) in 485 (83.8%) cases and benign disease (multinodular goiter, toxic diffuse goiter or toxic adenoma) in 94 (16.2%).

### 3.2. Surgical Procedure

Surgical procedure consisted in total thyroidectomy in 444 (76.7%) patients, in association with CLND in 110 (19%) and with modified radical lateral neck dissection in 25 (4.3%). The mean operative time was 97.1 ± 26.8 min, and the mean postoperative stay was 2.9 ± 1.1 days.

### 3.3. Pathological Examination

The mean nodule size was 13.7 ± 11.6 mm and the mean thyroid weight was 29.6 ± 24.2 g. Unexpected DTC was found in 94 (16.2%) cases. A microcarcinoma (T ≤ 1 cm) was found in 255 (44.1%) patients. Pathological examination revealed classic PTC in 288 (49.7%) cases, Follicular Variant of PTC (FV-PTC) in 168 (29%), diffuse sclerosing variant of PTC in 2 (0.3%), tall cell carcinoma in 43 (7.4%), FTC in 57 (9.8%) and Hürtle cell carcinoma (HCC) in 21 (3.6%). Multicentric tumor was found in 197 (34%) cases, extrathyroidal extension in 43 (7.4%) and angioinvasion in 19 (3.3%). Lymph node yield was 6.4 ± 8.2. Metastatic lymph nodes were found in 62 (10.7%) patients; the site of metastasis was central neck compartment in 40 cases, central and lateral neck compartment in 20 and only lateral compartment (skip metastasis) in 2. The mean lymph node ratio was 0.46 ± 0.29. 

### 3.4. Follow-Up

The mean follow-up was 55.1 ± 15.7 months (median 48, range 24–96). RAI therapy was administrated in 407 (70.3%) patients. A persistence or recurrence of DTC was found in 36 (6.2%) cases; specifically, we found a persistent disease in 10 patients and a recurrent disease in 26. The mean time between surgery and diagnosis of recurrence was 21.8 ± 14.2 months. The site of persistence or recurrence was central neck compartment in 25 (69.4%) cases, lateral neck compartment in 8 (22.2%), central and lateral neck compartment in 1 (2.8%) and lateral compartment and distant metastases in 2 (5.6%) (cerebral and adrenal metastases in one case and lung metastases in the other). Fourteen (38.9%) patients underwent subsequent reoperation, while the other 22 (66.1%) were treated with additional cycles of RAI ablation.

Among 289 patients with at least 5 years of follow-up, 5-year disease-free survival was 94.1%.

### 3.5. Statistical Analysis

Full univariate and multivariate analysis for preoperative data and surgical procedure are reported in Table 2, and for pathological features in Table 3.

#### 3.5.1. Univariate Analysis

Male sex was more frequent in patients with persistent or recurrent disease than in those disease-free (36.1% vs. 23.2%; *p* = 0.072); there were no other significant differences among demographic and preoperative factors between the two groups. Patients with recurrent or persistent disease had undergone more frequently lymphectomy than those in NED group (63.9% vs. 20.6%, considering both CLND and LND; *p* < 0.001). The nodule size and the thyroid weight were not significantly different between the two groups, but in case of microcarcinoma the incidence of recurrence was significantly lower (2.4% in patients with microcarcinoma vs. 9.2% in patients with T > 1 cm; *p* < 0.001). Histotype was associated with significantly different rate of recurrence (*p* < 0.01): patients with classic PTC and tall cell carcinoma were more likely to have a recurrence, whereas patients with FV-PTC and Hürtle cell carcinoma had a lower risk of recurrence. Lymph node yield (*p* < 0.01), lymph node metastases (*p* < 0.01), extrathyroidal invasion (*p* < 0.01), multicentricity (*p* < 0.01) and angioinvasion (*p* = 0.08) were significantly higher in patients with persistent or recurrent DTC.

#### 3.5.2. Multivariate and Survival Analysis

On multivariate analysis, only lymph node metastasis was identified as independent risk factor and microcarcinoma as independent protective factor for persistent or recurrent of disease. Specifically, the incidence of lymph node metastases was 44.4% in patients in the persistent or recurrent DTC group, and 8.3% in patients in the disease-free group (OR 4.274, *p* = 0.012), while the prevalence of microcarcinoma was 16% in the first group and 45.8% in the second (OR 0.328, *p* = 0.034).

A subset analysis was performed distinguishing between persistent and recurrent disease; significant results are reported in Table 4. When considering only recurrent disease, also angioinvasion (OR 6.181, *p* = 0.033), in addition to microcarcinoma (OR 0.218, *p* = 0.0472) and lymph node metastasis (OR 3.827, *p* = 0.0174), was found as independent risk factor. On the other hand, when including in analysis only patients with persistent DTC, only lymph node metastasis (OR 8.682, *p* = 0.0129) was identified as significant risk factor for persistent disease.

Kaplan-Meier curves for independent factors are reported in Figure 1a,b. Log-rank test demonstrated significant differences in survival curves both for lymph node metastases (*p* < 0.0001) and for microcarcinoma (*p* = 0.0304). Five-year disease-free survival was 95.8% in patients with negative lymph nodes and 73.7% in patients with metastatic lymph nodes, and 96.1% in patients with a microcarcinoma and 91.5% in patients with tumor >1 cm.

## 4. Discussion

Differentiated thyroid carcinoma is typically associated with a good prognosis, with a 20-year survival after surgery of over 90%. Nevertheless, some patients with DTC experience poor outcomes, with local or distant recurrence; in these patients, management is demanding, requiring a multidisciplinary evaluation including endocrinologist, surgeon, radiotherapist and oncologist to identify the proper treatment.

In this context, a suitable follow-up is essential to early identify patients with recurrence and to offer them the best therapy. In fact, an insufficient surveillance could lead to inadequate and delayed treatment; by the other side, it must be considered that an excess of unnecessary postoperative monitoring will have psychological and social negative impacts on patients.

In this regard, risk stratification is a process whose purpose is to identify groups of patients with homogenous features and similar prognosis; the goal of this procedure is to plan adequate treatment and follow-up for each group of patients, minimizing risks and maximizing benefits.

American Joint Committee on Cancer/Union for International Cancer Control (AJCC/UIC) staging system is a globally accepted classification of thyroid cancer which stratifies patients on the basis of age and histopathological findings, predicting disease-specific mortality. Nevertheless, this system is limited on factors that may influence disease-free survival, and then poorly helpful in the decision process regarding follow-up and treatment.

Over the years, several risk factors have been identified as independent predictors of prognosis, including age, gender, tumor size, histological grading and type, local or distant metastasis, extrathyroidal invasion and multicentricity. Based on these findings, different systems have been described to identify patients with high risk of recurrence.

AGES [11] system was described in 1987 and included age, grading, size, extrathyroidal invasion and distant metastasis; AMES [12] and MACIS [13] systems, reported in 1988 and 1993, were similar, also including male sex in AMES and incomplete surgical resection in MACIS. Lymph node metastases were considered as a risk factor only in the following systems, including OSU [9] in 1994, MSKCC [14] in 1995 and NTCTCS [15] in 1998.

The ATA Initial Risk Stratification System, purposed in 2009, is a three-tiered system in which patients are classified in low, medium and high-risk of persistent or recurrent disease based on clinicopathologic features; each category has a risk range for recurrence, which upper limit is 5% in low, 30% in intermediate and 55% in high-risk carcinomas. This system was enhanced in 2015 ATA guidelines (Modified Initial Risk Stratification System), presenting a continuum of risk of recurrence, ranging from <1% in very low risk patients to >50% in high-risk patients, based on factors such as histology, quality of resection during first surgery and, if available, genotype. For example, while all patients with an intrathyroidal DTC are considered in ATA low-risk category, irrespective of tumor size, the recurrence risk can vary from 1–2% in unifocal papillary microcarcinoma, to 4–6% in multifocal papillary microcarcinoma, to 8–10% in tumors with T > 4 cm. Similarly, patients with metastatic lymph nodes are classified in ATA intermediate-risk category, while the risk of recurrence can vary widely from 5% if less than 5 lymph nodes are involved, to 20% if more than 5 lymph nodes are involved, to 40% in patients with extranodal extension.

Furthermore, following a study of Tuttle [16] et al., ATA 2015 guidelines introduced a response-to-therapy variable, to take into account the outcome of treatment. Response to therapy can be classified in excellent when there is no evidence of disease (NED), biochemical incomplete response when there is no structural evidence of recurrence but abnormal Tg or Anti-Tg antibodies, and structural incomplete response when there is evidence of local or distant recurrence on imaging studies. Patients who fall in the category of structural incomplete response continue to have a persistent disease in 50–85% of cases despite additional therapy, and disease-specific mortality is 11% in patients with local metastases and 50% in patients with distant recurrence. The response-to-therapy variable is evaluated during the course of the disease, so that risk stratification becomes a dynamic process which may change over time due to clinical or laboratory findings.

These implementations reflect a modification in the approach to DTC, from a rigid and standardized management to individualized and tailored treatment and follow-up.

In our study the incidence of recurrence was 6.2%, with a 5-year disease-free survival of 94.1%. Our data are in the lower range of the recurrence rate reported in other studies, which varies widely from 2% to over 30%. In a long-term study on over 1300 patients with DTC, Mazzaferri reported a recurrence rate of 30% after 30 years [9], even if it should be noted that, since then, the abilities in diagnosis, management and follow-up has considerably improved, thus a lower recurrence rate should be expected.

Our study identified as independent factors lymph node metastasis, size of the tumor ≤1 cm and, when considering only recurrent disease, angioinvasion.

Bates [17] suggested that most of the patients classified as having a recurrence have indeed a persistent disease, meaning that the disease was already present at initial surgery but was misdiagnosed during preoperative work-up. In her work, only 3 cases were detected as “true recurrences”, while 71 reoperations were reclassified as persistent disease. Unfortunately, a comparative analysis between persistent and recurrent disease was not done due to the small number of patients in the recurrent group. The results of the present study confirm in part the findings of Bates: in our study we detected that about one third of the patients with relapse of disease had indeed a persistent disease. However, it must be noted that the two studies have a different research design: in fact, in the study of Bates were included only patients that had undergone a reoperation, while we included in our work also patients with persistent or recurrent disease that were managed non-operatively, and we compared these patients with those considered disease-free. Furthermore, we conducted an additional analysis separating persistent and recurrent disease, which results confirmed the influence of lymph node metastasis, angioinvasion and microcarcinoma on prognosis.

Lymph node metastasis has been largely described as a predictive factor for recurrence [18,19,20,21,22,23]. In our series, metastatic lymph nodes were present at initial surgery in 10.7% of patients. Patients with lymph node metastasis had a risk of recurrence 4 times higher than those with negative lymph nodes.

Recently, Kim [20] reported in a large study on 1928 patients with DTC without lateral neck metastases at diagnosis, a recurrence rate of 3.4% over a median follow-up period of 94 months; in his series, lateral neck recurrence was predicted by lymph node metastases in the central compartment, lymph node ratio and extranodal extension.

A lymph node ratio of 0.26 or higher was identified as independent risk factor of lateral neck recurrence in a multicentric study on 211 patients with PTC who underwent total thyroidectomy and bilateral prophylactic CLND [21]; a similar study reported a lymph node ratio >0.3 being associated with higher incidence of recurrence [24].

Raffaelli [22] identified lymph node yield, number of metastatic lymph nodes and lymph node ratio as predictive factor of recurrence on 209 patients who underwent LND.

In the present study, only lymph node metastasis, but not lymph node yield and lymph node ratio, was identified as independent predictive factor for recurrence. As stated also below, it is important to note that we included in the present work also patients that underwent surgery for benign disease and that had an incidental diagnosis of DTC; these tumors were unidentified at preoperative work-up and were almost all microcarcinomas, with a good prognosis. In these patients, a systematic lymphectomy was not performed and usually only a few perithyroidal lymph nodes were sampled and analyzed from the pathologist, thus it was sufficient that only one or two lymph nodes were positive to have a high lymph node ratio. This could explain why this variable was not significantly associated to higher risk of recurrence in our study.

By the other side, patients with aggressive tumors at clinical presentation, including those with lateral neck node metastases, had a more extensive surgery, and this can explain why patients who had lymph node dissection had, on univariate analysis, a significant higher incidence of recurrence.

Furthermore, 2015 ATA guidelines stated that if more than 5 lymph nodes are involved, the risk of recurrence is 20% and if extranodal invasion is present, the risk raises 40%; these findings are consistent with our work, where we had a recurrence rate of 47.4% in patients with more than 5 metastatic lymph nodes.

Since the beginning of the studies on DTC, the size of the carcinoma has been considered a factor influencing the prognosis. Microcarcinoma is defined as a tumor with largest diameter ≤1 cm and has extensively been reported as a positive prognostic factor, especially if unifocal and not associated with invasive features. ATA guidelines include this entity in low-risk category, with a risk of recurrence of 1–2%, not recommending postoperative RAI in patients who underwent thyroidectomy, and also suggesting active surveillance management as an alternative to surgery in patients with nodule ≤1 cm without evident metastasis or local invasion.

A Japanese study had reported more than a decade ago that thyroid microcarcinomas, followed without surgery for up to 10 years, tended to grow significantly only in 10.2% of patients, and that lymph node metastases appeared in only 1.2% of cases [25].

A recent study from Miyauchi [26] on 1211 low-risk papillary microcarcinoma submitted to active surveillance demonstrated that the probability of disease progression is related to the age at diagnosis: older patients have a significantly lower lifetime disease progression probability, being lower than 14.9% in patients older than 50 years; by the other side, younger patients have a higher lifetime risk of disease progression, being of 60.3% in patients in the 20s, 37.1% in 30 s and 27.3% in 40 s. This could induce to think that patients in their 20 s or 30 s have a not acceptable risk of disease progression to maintain active surveillance, requiring surgery; but on the other hand, 40% of patients in their 20 s and 63% of patients in their 30 s will not require a surgical treatment during their lifetime.

As already stated, in our series microcarcinoma was identified as an independent protective factor for recurrence. Among patients with microcarcinoma, we observed an overall disease-free survival of 97.6%; if considering only unifocal, non-invasive papillary microcarcinomas, we had only 2 cases of recurrence out of 163 cases, with disease-free survival of 98.8%.

Histologic variant of DTC is considered an important prognostic factor. ATA guidelines recognize as aggressive variants tall cell variant, columnar cell variant and hobnail variant; in these cases, if there are not high-risk features, the tumors are included in ATA intermediate-risk category, for which total thyroidectomy and postoperative RAI ablation are suggested.

Tall cell carcinoma is the most common aggressive variant of PTC, representing about 5–10% of PTC; it is usually diagnosed in older patients and is associated with aggressive pathologic features, with high rates of extrathyroidal extension and local and distant metastases [27,28,29,30,31]. In a recent meta-analysis on over 3000 PTC patients, Wang [32] reported a recurrence rate of 22% in tall cell carcinoma and 8.3% in classic PTC, with a triple incidence of distant metastases and a double rate of extrathyroidal extension in tall cell variant. Furthermore, tall cell variant of PTC frequently shows a decreased radioactive iodine uptake, being refractory to postoperative RAI ablation [33]. In our series, tall cell carcinoma was diagnosed in 43 patients with a mean age of 51.6 years; extrathyroidal invasion was present in 7 (16.3%) cases and lymph node metastases in 14 (32.6%). The incidence of recurrence was 13.9%, which was the highest rate among the different histotypes of DTC.

FV-PTC can be classified into two main subtypes: encapsulated and infiltrative. Encapsulated FV-PTC has been associated with a highly indolent clinical course in three different studies with a follow-up of at least 10 years. Liu [34] and Vivero [35] reported no recurrence in 61 and 27 patients respectively, and Baloch [36] reported 2 recurrences in 56 patients. Overall, the recurrence rate in these three studies was 1.4%. For these reasons, in 2016 a panel of endocrine experts suggested to rename this entity, removing the term “carcinoma” and advocating the definition of non-invasive follicular thyroid neoplasm with papillary-like nuclear features (NIFTP) [37]; this revision was accepted and adopted from World Health Organization in the following year [38]. According to classification used before 2017, when patients included in the current study underwent surgery, in this work we reported these tumors as encapsulated FV-PTC. Our findings are consistent with those found in literature: we had only one case of local recurrence out of 41 patients with encapsulated FV-PTC, with a recurrence rate of 2.4%. In the remnant 122 infiltrative subtype of FV-PTC, we found 4 (3.3%) recurrences.

HCC accounts for 3–4% of all DTC; it has been considered a variant of FTC until 2017, when Endocrine WHO decided to classify it as a separate histotype, due to differences in molecular and clinicopathologic features. HCC is usually diagnosed at an older age than FTC and is considered as having a worse prognosis than PTC and FTC. Nevertheless, the supposed greater aggressiveness of HCC remains still controversial: In fact, different studies have demonstrated that there is no significant difference in disease-free survival between patients with FTC and HCC, when adjusting for factors as age and tumor size [39,40]. Unexpectedly, in our series we had no recurrences on 21 cases of HCC.

Multifocality and bilateralism have been associated to increased risk of recurrence [41,42,43,44]. A recent study of Choi [45] on 2390 PTC demonstrated that multifocality is an independent predictive factor for recurrence, and that prognostic value of multifocality is significant in patients with PTC > 1 cm but not in microcarcinoma. These findings are in accordance with our study, where we found 197 cases of multicentric carcinoma, with a recurrence rate of 10.1%. All the recurrences occurred in tumors > 1 cm; excluding microcarcinoma, the recurrence rate was 12.9%.

Vascular invasion is defined as the invasion of vessels in the tumor capsule or beyond, with intravascular tumor cells attached to the vessel wall [46], even if stricter criteria have been proposed. Vascular invasion is considered an independent risk factor for recurrence. 2015 ATA guidelines include angioinvasive carcinomas in intermediate-risk group; PTC with vascular invasion has a risk of recurrence ranging from 15% to 30%. Furthermore, angioinvasion is an important feature in classifying FTC, which can be classified in minimally invasive (with only capsular invasion), encapsulated angioinvasive and widely invasive; these subtypes have different survival rate: 10-year survival has been reported as 98%, 80%, and 38%, respectively [47]. Similar results were found in the study by O’Neil [48]. ATA guidelines put on top of continuous risk scale FTC with extensive vascular invasion (defined as four or more foci of vascular invasion), with a risk of recurrence up to 55%.

Also, extrathyroidal extension (ETE) is deemed as independent risk factor for recurrence, presenting a risk up to 40%; tumors with gross ETE are classified in high-risk ATA category, requiring postoperative RAI and careful follow-up. ETE is strongly associated with distant metastases [49] and with aggressive features also in microcarcinomas [50]. ETE is often associated with BRAFV600E mutation, with a worse prognosis [51,52]. By the other side, it is important to note that the prognostic significance of minor ETE has been downsized from both ATA and AJCC system, because it does not affect disease-free and overall survival.

In our series, we reported 43 carcinomas with ETE: Among these, we had metastatic lymph nodes in 13 (30.2%) patients and we observed 7 (16.2%) cases of recurrence, with 2 patients having distant metastases.

As suggested in the 2015 ATA guidelines, to improve the accuracy of preoperative diagnosis, the use of cross-sectional imaging studies, such as CT and MRI, can be helpful as an adjunct to US for patients with clinical suspicion for advanced disease (including invasive primary tumor, or clinically apparent multiple or bulky lymph node involvement). In particular, regarding MRI, there is an increasing evidence that specific techniques, especially diffusion-weighted imaging, are able to distinguish more aggressive tumors from those less aggressive. This is a very important achievement that allows to choose the most appropriate treatment [53,54,55,56].

This study has several limitations. First, it is a retrospective analysis from a single institution, thus data may not be generalizable to all populations, also considering that ours is an endemic iodine-deficiency region, with a high incidence of autoimmune thyroiditis. Then, we included in the study also patients who underwent thyroidectomy for benign disease with incidental diagnosis of DTC after surgery: It’s likely that, in these patients, lymph nodes were under-sampled, thus the incidence of metastatic lymph nodes could be higher than that reported in our results. Finally, in our series we had a suboptimal follow-up time: Even if the recurrence usually presents at an early stage, there is a number of patients that present recurrence after a long time after surgery, thus it’s probably that in our study the real incidence of recurrence is underestimated.

## 5. Conclusions

DTC has generally a good prognosis. Tumors with larger diameter ≤1 cm have usually an indolent course, especially if not associated to other aggressive pathologic features: In this case, postoperative treatment and follow-up should be commensurate with the low-risk of recurrence. Further prospective studies are needed to clarify if non-operative management could be adequate in patients with microcarcinoma. Lymph node metastasis is an independent risk factor of recurrent disease: patients with this finding have mild to high-risk of recurrence, and postoperative therapy and follow-up should be aggressive. However, postoperative management should be tailored to the single patient, on the basis of clinical and pathological features and of response to therapy over time.

## Figures and Tables

**Figure 1 cancers-11-01230-f001:**
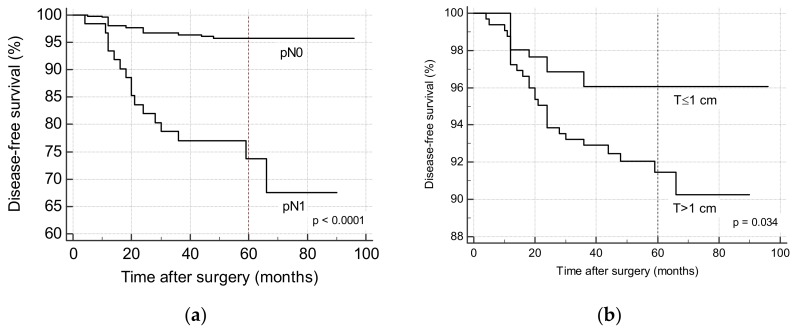
Kaplan-Meier curves estimating disease-free survival according to the presence of lymph node metastasis (**a**) and microcarcinoma (**b**).

**Table 1 cancers-11-01230-t001:** Clinical characteristics and surgical procedure of 579 patients analyzed in the study.

Variable	Patients (*n* = 579)
Sex	
- Male	139 (24%)
- Female	440 (76%)
Age (years)	50.7 ± 14.1
Familiarity for thyroid carcinoma	103 (17.8%)
Hyperthyroidism	53 (9.2%)
Autoimmune thyroiditis	215 (21.6%)
Surgical procedure	
- TT	444 (76.7%)
- TT + CLND	110 (19%)
- TT + CLND + LND	25 (4.3%)
Nodule size (mm)	13.7 ± 11.6
Microcarcinoma	255 (44.1%)
Lymph node metastases	62 (10.7%)
Follow up (months)	55 ± 15.7
NED	543 (93.8%)
Persistent/recurrent disease	36 (6.2%)

TT: Total Thyroidectomy; CLND: Central Lymph Node Dissection; LND: Lateral Neck Dissection.

**Table 2 cancers-11-01230-t002:** Univariate and multivariate analysis of preoperative data and surgical procedure of 579 patients with differentiated thyroid carcinoma.

Variable	Univariate Analysis			Multivariate Analysis			
	NED (*n* = 543)	Persistent/Recurrent Disease (*n* = 36)	*p*-Value	Regression Coefficient	Odds Ratio	95% CI	*p*-Value
Male sex	126 (23.2%)	13 (36.1%)	**0.072**	0.4867	1.626	0.734–3.605	0.231
Age (years)	50.8 ± 14.1	49.8 ± 13.3	0.68				
Age > 40 years	402 (74%)	26 (72%)	0.81				
Familiarity for thyroid carcinoma	13 (2.4%)	1 (2.8%)	0.88				
Hyperthyroidism	55 (10.1%)	1 (2.8%)	0.14				
Autoimmune thyroiditis	214 (39.4%)	13 (36.1%)	0.69				
Surgical procedure			***p* < 0.001**				
- TT	431 (79.4%)	13 (36.1%)		1.000	1.000	Reference	
- TT + CLND	96 (17.7%)	14 (38.9%)		−0.602	0.548	0.159–1.878	0.339
- TT + CLND + LND	16 (2.9%)	9 (25%)		0.592	1.807	0.297–10.998	0.521

NED: No Evidence of Disease; TT: Total Thyroidectomy; CLND: Central Lymph Node Dissection; LND: Lateral Neck Dissection. (*p*-values highlighted in bold are to be considered statistically significant).

**Table 3 cancers-11-01230-t003:** Univariate and multivariate analysis of pathological data of 579 patients with differentiated thyroid carcinoma.

Variable	Univariate Analysis			Multivariate Analysis			
	NED (*n* = 543)	Persistent/Recurrent Disease (*n* = 36)	*p*-Value	Regression Coefficient	Odds Ratio	95% CI	*p*-Value
Nodule size	13.5 ± 11.5	16.1 ± 12.1	0.21				
Microcarcinoma	249 (45.8%)	6 (16.6%)	***p* < 0.001**	−1.114	0.328	0.117–0.920	**0.034**
Thyroid weight	28.2 ± 19	29.7 ± 24.6	0.72				
Histotype			***p* < 0.01**				
- PTC	267 (49.2%)	21 (58.3%)		1.000	1.000	Reference	
- FV−PTC	163 (30%)	5 (13.9%)		−0.721	0.486	0.166–1.425	0.189
- Tall cell carcinoma	37 (6.8%)	6 (16.7%)		0.209	1.233	0.397–3.826	0.716
- Diffuse sclerosing variant of PTC	1 (0.2%)	1 (2.8%)		1.476	4.374	0.202–94.541	0.347
- FTC	54 (9.9%)	3 (8.3%)		−0.474	0.622	0.154–2.514	0.505
- Hürtle cell carcinoma	21 (3.9%)	0		−19.818	<0.001		0.998
Lymph node yield	5.8 ± 7.7	14 ± 10.7	***p* < 0.01**	−0.012	0.988	0.929–1.049	0.697
Lymph node metastasis	45 (8.3%)	16 (44.4%)	***p* < 0.01**	1.453	4.274	1.367–13.359	**0.012**
Lymph node ratio	0.44 ± 0.29	0.5 ± 0.28	0.53				
Extrathyroidal invasion	36 (6.6%)	7 (19.4%)	***p* < 0.01**	0.258	1.295	0.444–3.775	0.636
Multicentric carcinoma	177 (32.6%)	20 (55.6%)	***p* < 0.01**	−1.114	1.423	0.632–3.201	0.394
Angioinvasive carcinoma	16 (2.9%)	3 (8.3%)	**0.08**	0.959	2.611	0.567–12.050	0.219

NED: No Evidence of Disease; PTC: Papillary Thyroid Carcinoma; FV-PTC: Follicular variant of PTC; FTC: Follicular Thyroid Carcinoma. (*p*-values highlighted in bold are to be considered statistically significant).

**Table 4 cancers-11-01230-t004:** Report of significant independent factors at multivariate analysis for persistent and recurrent disease.

Variable	Persistent Disease (*n* = 10)				Recurrent Disease (*n* = 26)			
	Regression Coefficient	Odds Ratio	95% CI	*p*-Value	Regression Coefficient	Odds Ratio	95% CI	*p*-Value
Microcarcinoma	−0.37710	0.6859	0.1041–4.5179	0.6950	−1.51926	0.2189	0.0488–0.9812	**0.0472**
Lymph node metastasis	2.16126	8.6821	1.5796 to 47.7213	**0.0129**	1.34217	3.8273	1.2665–11.5661	**0.0174**
Angioinvasive carcinoma	−18.64235	<0.001	-	0.9986	1.82154	6.1813	1.1587–32.9760	**0.0330**

(*p*-values highlighted in bold are to be considered statistically significant).

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
