# Peer review of "Predictive Factors of Recurrence in Patients with Differentiated Thyroid Carcinoma: A Retrospective Analysis on 579 Patients"

_cancers, 2019, doi:10.3390/cancers11091230_

Round 1
Reviewer 1 Report
Dear authors,
this is a very nice retrospective analysis of a large single center database on DTCs. The study nicely demonstrates factors associated with recurrence of this relatively benign malignancy.
I recommend to adjust the following minor points prior acceptance for publication.
1) For my understanding, serum tumor markers were not included in the regression analysis. Please amend. It would be very important to see whether the disease status indeed can be monitored using the classical serum markers.
2) Did some of the patients underwent MRI prior surgery? There is an increasing body of evidence that specific MRI techniques, especially diffusion weighted imaging, are able to distinguish more aggressive phenotypes from less aggressive (metastatic vs. non-met., invasive vs. confined etc). This is a very important point, as non-invasive presurgical imaging offering this level of information should be implemented in patient care and used for identification of the most appropriate therapy. Therefore, please add the following references in your discussion: doi: 10.1016/j.tranon.2016.09.001; 10.3390/ijms18040821; 10.1371/journal.pone.0200270. A further study even demonstrated the predictive value of DWI for metastatic lymphatic dissemination, although not yet in thyroid malignancies 10.1007/s11307-017-1073-y.
3) The discussion definitely needs more focus. It has too much of a review character in the current form. Please shorten and focus appropriately. In my opinion, everything important can be said within 1.5 - 2 pages.
Author Response
Dear reviewer, thank you very much for your comments and suggestions.
1) For my understanding, serum tumor markers were not included in the regression analysis. Please amend. It would be very important to see whether the disease status indeed can be monitored using the classical serum markers.
I agree with your observation but, unfortunately, the patients were identified retrospectively from a prospectively maintained database; in this database, especially for the initial years, only qualitative information (positivity/negativity) were reported regarding serum tumor markers, thus full quantitative data is missing, precluding univariate and multivariate analysis.
2) Did some of the patients underwent MRI prior surgery? There is an increasing body of evidence that specific MRI techniques, especially diffusion weighted imaging, are able to distinguish more aggressive phenotypes from less aggressive (metastatic vs. non-met., invasive vs. confined etc). This is a very important point, as non-invasive presurgical imaging offering this level of information should be implemented in patient care and used for identification of the most appropriate therapy. Therefore, please add the following references in your discussion: doi: 10.1016/j.tranon.2016.09.001; 10.3390/ijms18040821; 10.1371/journal.pone.0200270. A further study even demonstrated the predictive value of DWI for metastatic lymphatic dissemination, although not yet in thyroid malignancies 10.1007/s11307-017-1073-y.
I agree with your suggestion, in fact, currently, we perform MRI or CT as an adjunct to US for patients with clinical suspicion for advanced disease, as suggested in the 2015 ATA guidelines. However, when patients analyzed in this study underwent surgery (between 2011 and 2016) this recommendation was not yet available and therefore MRI or CT was not systematically performed in case of advanced malignancy. For this reason, in the first version of the manuscript, we did not mention this radiological examination. However, we have added a brief comment in the "discussion" section and the references you suggested.
3) The discussion definitely needs more focus. It has too much of a review character in the current form. Please shorten and focus appropriately. In my opinion, everything important can be said within 1.5 - 2 pages.
I've shortened it, but then it became longer because I had to specify several points following your and other reviewer's comments and suggestions.
Reviewer 2 Report
This manuscript describes a restrospective study on a cohort of DTC patients, with a focus on identifying clinical or pathological factors associated with biochemical or structural recurrence. Based on multivariate analyses, the presence of lymph node metastases is a factor positively associated with recurrence, and a diagnosis of microcarcinoma is a factor negatively associated with recurrence.
Major comments
1. My main concern about this study, is that there is no distinction made between “recurrent” disease and “residual” disease. In fact, studies have shown that most “recurrent” disease discovered after the initial treatment of DTC is actually persistent disease (i.e., residual disease), that was already present, but not detected during the preoperative work-up (Bates MF et al. 2018 Surgery; https://www.ncbi.nlm.nih.gov/pubmed/29128176). It is therefore important to make this distinction. In its current form, the study addresses both recurrent and residual disease together, not distinguishing between the two, and essentially pooling together all patients with evidence of disease after the initial treatment. I recommend to perform an additional analysis, separating the two categories of patients: (i) Those who achieved a “no evidence of disease” (NED) status at some point after the initial treatment, and then were later found to have biochemical or structural evidence of disease (this would be the “real recurrent disease” group); and (ii) those who never achieved a NED status (this would be the “residual disease” group). The study cited above could help as reference to design this analysis. Also, for the disease-free survival curves of this particular analysis, it would be important to assess each patient not from the time of the initial surgery (as is presently done in the manuscript), but from the moment when they were declared disease-free. It will then be interesting to see whether factor emerge that can identify risk for “real” recurrence after a NED status. It will also be interesting to see whether the present study agrees or not with the findings of the paper cited above.
2. It should be stated in the manuscript whether the study received ethical approval from a competent body. If no such approval is required for this type of study in Italy, then this should be stated instead.
Minor comments
3. The manuscript uses “recurrence” and “relapse” interchangeably, and this can lead to confusion. I recommend to use only one or the other term, as they mean the same thing.
4. In Table 1, please specify if values shown mean or median, and if the plus/minus values show SME or SD.
5. In section 2.4, please rephrase to avoid saying that p-values y0.1 were considered “significant”; it would be better to say something like “potentially relevant”.
6. In section 3.3, please mention NIFTP, and explain whether it was included in the FV-PTC category for this study.
7. In Table 3, the p-value of 0.012 for lymph nodes should be in bold.
8. In section 3.5.1, please discuss the fact that patients who had lymph node dissection also had more “recurrence”. Obviously, it is because they had lymph node metastases to begin with, and not because they had more extensive surgery. It is obvious for experienced readers, but it would still be nice to state it formally.
9. In line 238, please consider that the Mazzaferri study reporting a high rate of DTC recurrence was published in 1994, reviewing a period of 30 years in the past, and please discuss that our abilities to diagnose, manage and follow DTC have improved very much since then; therefore, much lower recurrence rates should be currently expected.
10. Since the Discussion is very complete, please mention also in the section about extra-thyroidal extension (ETE) that the current AJCC system (not used in the present study) has down-staged minimal ETE because it does not affect survival.
Author Response
Dear Reviewer, thank you very much for your comments.
1. My main concern about this study, is that there is no distinction made between “recurrent” disease and “residual” disease. In fact, studies have shown that most “recurrent” disease discovered after the initial treatment of DTC is actually persistent disease (i.e., residual disease), that was already present, but not detected during the preoperative work-up (Bates MF et al. 2018 Surgery; https://www.ncbi.nlm.nih.gov/pubmed/29128176). It is therefore important to make this distinction. In its current form, the study addresses both recurrent and residual disease together, not distinguishing between the two, and essentially pooling together all patients with evidence of disease after the initial treatment. I recommend to perform an additional analysis, separating the two categories of patients: (i) Those who achieved a “no evidence of disease” (NED) status at some point after the initial treatment, and then were later found to have biochemical or structural evidence of disease (this would be the “real recurrent disease” group); and (ii) those who never achieved a NED status (this would be the “residual disease” group). The study cited above could help as reference to design this analysis. Also, for the disease-free survival curves of this particular analysis, it would be important to assess each patient not from the time of the initial surgery (as is presently done in the manuscript), but from the moment when they were declared disease-free. It will then be interesting to see whether factor emerge that can identify risk for “real” recurrence after a NED status. It will also be interesting to see whether the present study agrees or not with the findings of the paper cited above.
This is a very important suggestion. I’ve added in Methods the distinction between persistent and recurrent disease (lines 95-98). Then, following your recommendation, I’ve done an additional analysis separating patients with persistent disease and patients with recurrent disease (lines 180-185). Indeed, about 1/3 of patients had persistent disease. The results of the additional analysis are very interesting and reinforce previous findings: in fact in the recurrent group also angioinvasion, other than lymph node metastasis and microcarcinoma, was found as an independent risk factor. By the other side, in persistent group only lymph node metastasis was significant.
I decided to report in a new table (Table 4) only significant factors to avoid too many long tables in the paper.
The results of our study are compared to those found from Bates and colleagues in Discussion (lines 260-273).
Also, for the disease-free survival curves of this particular analysis, it would be important to assess each patient not from the time of the initial surgery (as is presently done in the manuscript), but from the moment when they were declared disease-free. It will then be interesting to see whether factor emerge that can identify risk for “real” recurrence after a NED status. It will also be interesting to see whether the present study agrees or not with the findings of the paper cited above.
Thank you very much for this observation. Unfortunately, the time needed to “NED status” declaration is not uniform and well defined, as also stated in the article you cited; for example, patients who need postoperative RAI therapy will wait 3-6 months (and in some cases up to one year) just depending on waiting list for RAI therapy, and then will have a late in NED classification when compared to patients who don’t need RAI therapy, that will be declared as NED earlier. Thus, I’m afraid that changing initial time of survival curve from operation to NED status declaration could be a bias and a confounding factor. However, following your observation, I’ve tried to re-calculate Kaplan-Meier curves differentiating recurrent and persistent disease (see attached file). Results are consistent with the additional multivariate analysis already discussed. I preferred to leave the two original figures, including persistent and recurrent disease together, to not have too many images in the paper.
2. It should be stated in the manuscript whether the study received ethical approval from a competent body. If no such approval is required for this type of study in Italy, then this should be stated instead.
I've stated it in Methods (lines 53-55).
3. The manuscript uses “recurrence” and “relapse” interchangeably, and this can lead to confusion. I recommend to use only one or the other term, as they mean the same thing.
Done.
4. In Table 1, please specify if values shown mean or median, and if the plus/minus values show SME or SD.
Considering that continuous variables are present also in the other tables, I’ve added a sentence at the end of Methods specifying that Continuous variables are expressed as mean ± standard deviation of the mean.
5. In section 2.4, please rephrase to avoid saying that p-values y0.1 were considered “significant”; it would be better to say something like “potentially relevant”.
Done.
6. In section 3.3, please mention NIFTP, and explain whether it was included in the FV-PTC category for this study.
This paper refers to patients that underwent surgery before 2017, when the new classification of NIFTP was introduced and accepted from WHO. For this reason, we decided to report these tumors as follicular variant of PTC, respecting original classification, and not as NIFPT, also considering that the specimens were not reviewed from the pathologist for the purpose of this study. I’ve specified it in the Discussion (lines 341-350).
7. In Table 3, the p-value of 0.012 for lymph nodes should be in bold.
Done.
8. In section 3.5.1, please discuss the fact that patients who had lymph node dissection also had more “recurrence”. Obviously, it is because they had lymph node metastases to begin with, and not because they had more extensive surgery. It is obvious for experienced readers, but it would still be nice to state it formally.
I’ve underlined this aspect in Discussion (lines 297-299).
9. In line 238, please consider that the Mazzaferri study reporting a high rate of DTC recurrence was published in 1994, reviewing a period of 30 years in the past, and please discuss that our abilities to diagnose, manage and follow DTC have improved very much since then; therefore, much lower recurrence rates should be currently expected.
Done (lines 255-257).
10. Since the Discussion is very complete, please mention also in the section about extra-thyroidal extension (ETE) that the current AJCC system (not used in the present study) has down-staged minimal ETE because it does not affect survival.
Done (383-385)

Reviewer 3 Report
In this interesting manuscript, the authors explored the indipendent predictive factors of recurrence in patients who underwent thyroidectomy for differentiated thyroid cancers (DTCs). The increasing incidence of DTCs detected in the last decades worldwide as well as the not negligible possibility of local and distant relapse of disease, make a postoperative individualized management and an adequate surveillance of the patient extremely important. Overall, the manuscript is well written, introduction contains sufficient elements for background, and results have been extensively discussed.
MINOR REVISIONS
Methods:
Line 68: Consider to write full names for ATA and NCNN.
Line 70: The reference quoted at line 70 should be reported in bibliography.
Results:
Paragraph 3.3 Consider to include these descriptive results within Table 1.
Line 126. Is FC the abbreviation of Follicular Thyroid Cancer? The authors used the acronym FTC and not FC at line 32.
Table 2. The percentage of PTC cases with recurrent disease and with no evidence of disease are missing.
Line 158. The percentage of patients who underwent only thyroidectomy seems incorrect if compared to the value reported in table 2.
Line 161. Based on my computations, the correct incidence of recurrence in patients with tumor size >1 cm is 9.2% and not 8%.
Line 174. Please replace “Figure 1” and “Figure 2” with “Figure 1a” and “Figure 1b”, respectively.
Discussion:
Lines 205-209. Consider to write full names of the systems quoted.
Line 242. The value reported (10.5%) is different from that at line 129 (10.7%).
Lines 250-251. The limph node ratio observed in the study does not appear to be a predictive factor of relapse (Table 4). Could you discuss this specific point?
Line 256. …“recurrence rate of 47.4% in patients with more than 5 metastatic lymph nodes”. I was not able to find this information in table 4.
Lies 301-302. Total cases of FVPTC, as reported in table 4, are 163, and 5 of them are patients with recurrence of disease. Please explain this apparent difference from what you stated in discussion.
Author Response
Dear reviewer, thank you very much for your comments.
Methods:
Line 68: Consider to write full names for ATA and NCNN.
Done.
Line 70: The reference quoted at line 70 should be reported in bibliography.
Done.
Results:
Paragraph 3.3 Consider to include these descriptive results within Table 1.
We decided to not include these results in Table 1 because they are extensively reported in Tables 2 and 3, even if divided in NED vs persistent or recurrent disease. The purpose of the first table was to be an “eye-catcher” on the series. However, following your suggestion, I’ve added in Table 1 some essential results as microcarcinoma, lymph node metastases and persistent or recurrent disease.
Line 126. Is FC the abbreviation of Follicular Thyroid Cancer? The authors used the acronym FTC and not FC at line 32.
Done.
Table 2. The percentage of PTC cases with recurrent disease and with no evidence of disease are missing.
Following your comment, I’ve added the number and percentage of patients with NED and with recurrence in Table 1. We preferred to not add the percentage in Table 2 and Table 3 to avoid confusion, because NED and recurrent disease in these tables indicate the total of each subgroup and to which the subsequent independent factors (the rows of the tables) are referred.
Line 158. The percentage of patients who underwent only thyroidectomy seems incorrect if compared to the value reported in table 2.
The original sentence was unclear, I’ve changed it (lines 164-166).
Line 161. Based on my computations, the correct incidence of recurrence in patients with tumor size >1 cm is 9.2% and not 8%.
Your computation is right, I've changed it.
Line 174. Please replace “Figure 1” and “Figure 2” with “Figure 1a” and “Figure 1b”, respectively.
Done.
Discussion:
Lines 205-209. Consider to write full names of the systems quoted.
I would prefer to maintain the acronyms because it's explained in the same sentences what do they refer to.
Line 242. The value reported (10.5%) is different from that at line 129 (10.7%).
I've changed it.
Lines 250-251. The limph node ratio observed in the study does not appear to be a predictive factor of relapse (Table 4). Could you discuss this specific point?
This is a very interesting suggestion. I’ve added this sentences in the discussion: “It is important to note that we included in the present work also patients that underwent surgery for benign disease and that had an incidental diagnosis of DTC; these tumors were almost all microcarcinomas, unidentified at preoperative US, with a good prognosis. In these patients, a systematic lymphectomy was not performed and usually only a few perithyroidal lymph nodes were sampled and analyzed from the pathologist, thus it was sufficient that only one or two lymph nodes were positive to have a high lymph node ratio. This could explain why this variable was not significantly associated to higher risk of recurrence in our study".
Line 256. …“recurrence rate of 47.4% in patients with more than 5 metastatic lymph nodes”. I was not able to find this information in table 4.
We decided to not include this variable (>5 positive lymph nodes) in multivariate analysis to reduce the risk of overfitting the model due to the presence of too many similar independent variables (in this case, we already had lymph node metastasis, lymph node ratio and lymph node yield that are strictly related to each other). This is the reason why we didn’t enter this information in table 3, but we thought that it was interesting to cite it during the discussion because it is one of the risk factors identified from ATA.
Lies 301-302. Total cases of FVPTC, as reported in table 4, are 163, and 5 of them are patients with recurrence of disease. Please explain this apparent difference from what you stated in discussion.
I’ve modified the entire paragraph that was unclear, specifying that there are two subgroups of FV-PTC (encapsulated and infiltrative), and that the one with a very good prognosis is the encapsulated variant, now classified as NIFPT. I’ve specified at the end of the paragraph that, among a total of 163 FVPTC, we had 41 encapsulated FV-PTC (with only one recurrence) and 122 infiltrative PV-PTC (with 4 recurrences).
Round 2
Reviewer 2 Report
I commend the authors for having thoroughly revised the manuscript according to the comments in the first review round. I have no further comments.